# Structured Estimation with Atomic Norms: General Bounds and Applications

**Sheng Chen**          **Arindam Banerjee**
Dept. of Computer Science & Engg., University of Minnesota, Twin Cities
{shengc,banerjee}@cs.umn.edu

## Abstract

For structured estimation problems with atomic norms, recent advances in the literature express sample complexity and estimation error bounds in terms of certain *geometric measures*, in particular Gaussian width of the unit norm ball, Gaussian width of a spherical cap induced by a tangent cone, and a restricted norm compatibility constant. However, given an atomic norm, bounding these geometric measures can be difficult. In this paper, we present general upper bounds for such geometric measures, which only require simple information of the atomic norm under consideration, and we establish tightness of these bounds by providing the corresponding lower bounds. We show applications of our analysis to certain atomic norms, especially $k$-support norm, for which existing result is incomplete.

## 1   Introduction

Accurate recovery of structured sparse signal/parameter vectors from noisy linear measurements has been extensively studied in the field of compressed sensing, statistics, etc. The goal is to recover a high-dimensional signal (parameter) $\boldsymbol{\theta}^* \in \mathbb{R}^p$ which is sparse (only has a few nonzero entries), possibly with additional structure such as group sparsity. Typically one assume linear models, $\mathbf{y} = \mathbf{X}\boldsymbol{\theta}^* + \boldsymbol{\omega}$, in which $\mathbf{X} \in \mathbb{R}^{n \times p}$ is the design matrix consisting of $n$ samples, $\mathbf{y} \in \mathbb{R}^n$ is the observed response vector, and $\boldsymbol{\omega} \in \mathbb{R}^n$ is an unknown noise vector. By leveraging the sparsity of $\boldsymbol{\theta}^*$, previous work has shown that certain $L_1$-norm based estimators [22, 7, 8] can find a good approximation of $\boldsymbol{\theta}^*$ using sample size $n \ll p$. Recent work has extended the notion of unstructured sparsity to other structures in $\boldsymbol{\theta}^*$ which can be captured or approximated by some norm $R(\cdot)$ [10, 18, 3, 11, 6, 19] other than $L_1$, e.g., (non)overlapping group sparsity with $L_1/L_2$ norm [24, 15], etc. In general, two broad classes of estimators are considered in recovery analysis: (i) Lasso-type estimators [22, 18, 3], which solve the regularized optimization problem

$$\hat{\boldsymbol{\theta}}_{\lambda_n} = \operatorname*{argmin}_{\boldsymbol{\theta} \in \mathbb{R}^p} \frac{1}{2n} \|\mathbf{X}\boldsymbol{\theta} - \mathbf{y}\|_2^2 + \lambda_n R(\boldsymbol{\theta}) \;, \tag{1}$$

and (ii) Dantzig-type estimators [7, 11, 6], which solve the constrained problem

$$\hat{\boldsymbol{\theta}}_{\lambda_n} = \operatorname*{argmin}_{\boldsymbol{\theta} \in \mathbb{R}^p} \; R(\boldsymbol{\theta}) \quad \text{s.t.} \quad R^*(\mathbf{X}^T(\mathbf{X}\boldsymbol{\theta} - \mathbf{y})) \leq \lambda_n \;, \tag{2}$$

where $R^*(\cdot)$ is the dual norm of $R(\cdot)$. Variants of these estimators exist [10, 19, 23], but the recovery analysis proceeds along similar lines as these two classes of estimators.

To establish recovery guarantees, [18] focused on Lasso-type estimators and $R(\cdot)$ from the class of *decomposable* norm, e.g., $L_1$, non-overlapping $L_1/L_2$ norm. The upper bound for the estimation error $\|\hat{\boldsymbol{\theta}}_{\lambda_n} - \boldsymbol{\theta}^*\|_2$ for any decomposable norm is characterized in terms of three *geometric measures*: (i) a dual norm bound, as an upper bound for $R^*(\mathbf{X}^T\boldsymbol{\omega})$, (ii) sample complexity, the minimal sample size needed for a certain *restricted eigenvalue (RE)* condition to be true [4, 18], and (iii) a *restricted*

*norm compatibility* constant between $R(\cdot)$ and $L_2$ norms [18, 3]. The non-asymptotic estimation error bound typically has the form $\|\hat{\boldsymbol{\theta}}_{\lambda_n} - \boldsymbol{\theta}^*\|_2 \leq c/\sqrt{n}$, where $c$ depends on a product of dual norm bound and restricted norm compatibility, whereas the sample complexity characterizes the minimum number of samples after which the error bound starts to be valid. In recent work, [3] extended the analysis of Lasso-type estimator for decomposable norm to any norm, and gave a more succinct characterization of the dual norm bound for $R^*(\mathbf{X}^T\boldsymbol{\omega})$ and the sample complexity for the RE condition in terms of *Gaussian widths* [14, 10, 20, 1] of suitable sets where, for any set $\mathcal{A} \in \mathbb{R}^p$, the Gaussian width is defined as

$$w(\mathcal{A}) = \mathbb{E} \sup_{\mathbf{u} \in \mathcal{A}} \langle \mathbf{u}, \mathbf{g} \rangle \ , \tag{3}$$

where $\mathbf{g}$ is a standard Gaussian random vector. For Dantzig-type estimators, [11, 6] obtained similar extensions. To be specific, assume entries in $\mathbf{X}$ and $\boldsymbol{\omega}$ are i.i.d. normal, and define the *tangent cone*,

$$\mathcal{T}_R(\boldsymbol{\theta}^*) = \text{cone} \left\{ \mathbf{u} \in \mathbb{R}^p \mid R(\boldsymbol{\theta}^* + \mathbf{u}) \leq R(\boldsymbol{\theta}^*) \right\} \ . \tag{4}$$

Then one can get (high-probability) upper bound for $R^*(\mathbf{X}^T\boldsymbol{\omega})$ as $O(\sqrt{n}w(\Omega_R))$ where $\Omega_R = \{\mathbf{u} \in \mathbb{R}^p | R(\mathbf{u}) \leq 1\}$ is the *unit norm ball*, and the RE condition is satisfied with $O(w^2(\mathcal{T}_R(\boldsymbol{\theta}^*) \cap \mathbb{S}^{p-1}))$ samples, in which $\mathbb{S}^{p-1}$ is the unit sphere. For convenience, we denote by $\mathcal{C}_R(\boldsymbol{\theta}^*)$ the spherical cap $\mathcal{T}_R(\boldsymbol{\theta}^*) \cap \mathbb{S}^{p-1}$ throughout the paper. Further, the restricted norm compatibility is given by $\Psi_R(\boldsymbol{\theta}^*) = \sup_{\mathbf{u} \in \mathcal{T}_R(\boldsymbol{\theta}^*)} \frac{R(\mathbf{u})}{\|\mathbf{u}\|_2}$ (see Section 2 for details).

Thus, for any given norm, it suffices to get a characterization of (i) $w(\Omega_R)$, the width of the unit norm ball, (ii) $w(\mathcal{C}_R(\boldsymbol{\theta}^*))$, the width of the spherical cap induced by the tangent cone $\mathcal{T}_R(\boldsymbol{\theta}^*)$, and (iii) $\Psi_R(\boldsymbol{\theta}^*)$, the restricted norm compatibility in the tangent cone. For the special case of $L_1$ norm, accurate characterization of all three measures exist [10, 18]. However, for more general norms, the literature is rather limited. For $w(\Omega_R)$, the characterization is often reduced to comparison with either $w(\mathcal{C}_R(\boldsymbol{\theta}^*))$ [3] or known results on other norm balls [13]. While $w(\mathcal{C}_R(\boldsymbol{\theta}^*))$ has been investigated for certain decomposable norms [10, 9, 1], little is known about general non-decomposable norms. One general approach for upper bounding $w(\mathcal{C}_R(\boldsymbol{\theta}^*))$ is via the *statistical dimension* [10, 19, 1], which computes the expected squared distance between a Gaussian random vector and the *polar cone* of $\mathcal{T}_R(\boldsymbol{\theta}^*)$. To specify the polar, one need full information of the subdifferential $\partial R(\boldsymbol{\theta}^*)$, which could be difficult to obtain for non-decomposable norms. A notable bound for (overlapping) $L_1/L_2$ norms is presented in [21], which yields tight bounds for mildly non-overlapping cases, but is loose for highly overlapping ones. For $\Psi_R(\boldsymbol{\theta}^*)$, the restricted norm compatibility, results are only available for decomposable norms [18, 3].

In this paper, we present a general set of bounds for the width $w(\Omega_R)$ of the norm ball, the width $w(\mathcal{C}_R(\boldsymbol{\theta}^*))$ of the spherical cap, and the restricted norm compatibility $\Psi_R(\boldsymbol{\theta}^*)$. For the analysis, we consider the class of atomic norms that are *invariant under sign-changes*, i.e., the norm of a vector stays unchanged if any entry changes only by flipping its sign. The class is quite general, and covers most of the popular norms used in practical applications, e.g., $L_1$ norm, ordered weighted $L_1$ (OWL) norm [5] and $k$-support norm [2]. Specifically we show that sharp bounds on $w(\Omega_R)$ can be obtained using simple calculation based on a decomposition inequality from [16]. To upper bound $w(\mathcal{C}_R(\boldsymbol{\theta}^*))$ and $\Psi_R(\boldsymbol{\theta}^*)$, instead of a full specification of $\mathcal{T}_R(\boldsymbol{\theta}^*)$, we only require some information regarding the subgradient of $R(\boldsymbol{\theta}^*)$, which is often readily accessible. The key insight is that bounding statistical dimension often ends up computing the expected distance from Gaussian vector to a single point rather than to the whole polar cone, thus the full information on $\partial R(\boldsymbol{\theta}^*)$ is unnecessary. In addition, we derive the corresponding lower bounds to show the tightness of our results. As examples, we illustrate the bounds for $L_1$ and OWL norms [5]. Finally, we give sharp bounds for the recently proposed $k$-support norm [2], for which existing analysis is incomplete.

The rest of the paper is organized as follows: we first review the relevant background for Dantzig-type estimator and atomic norm in Section 2. In Section 3, we introduce the general bounds for the geometric measures. In Section 4, we discuss the tightness of our bounds. Section 5 is dedicated to the example of $k$-support norm, and we conclude in Section 6.

## 2  Background

In this section, we briefly review the recovery guarantee for the *generalized Dantzig selector* in (2) and the basics on atomic norms. The following lemma, originally [11, Theorem 1], provides an error bound for $\|\hat{\boldsymbol{\theta}}_{\lambda_n} - \boldsymbol{\theta}^*\|_2$. Related results have appeared for other estimators [18, 10, 19, 3, 23].

**Lemma 1** *Assume that* $\mathbf{y} = \mathbf{X}\boldsymbol{\theta}^* + \boldsymbol{\omega}$, *where entries of* $\mathbf{X}$ *and* $\boldsymbol{\omega}$ *are i.i.d. copies of standard Gaussian random variable. If* $\lambda_n \geq c_1\sqrt{n}w(\Omega_R)$ *and* $n > c_2 w^2(\mathcal{T}_R(\boldsymbol{\theta}^*) \cap \mathbb{S}^{p-1}) = w^2(\mathcal{C}_R(\boldsymbol{\theta}^*))$ *for some constant* $c_1, c_2 > 1$, *with high probability, the estimate* $\hat{\boldsymbol{\theta}}_{\lambda_n}$ *given by* (2) *satisfies*

$$\|\hat{\boldsymbol{\theta}}_{\lambda_n} - \boldsymbol{\theta}^*\|_2 \leq O\left(\Psi_R(\boldsymbol{\theta}^*)\frac{w(\Omega_R)}{\sqrt{n}}\right) . \tag{5}$$

In this Lemma, there are three geometric measures—$w(\Omega_R)$, $w(\mathcal{C}_R(\boldsymbol{\theta}^*))$ and $\Psi_R(\boldsymbol{\theta}^*)$—which need to be determined for specific $R(\cdot)$ and $\boldsymbol{\theta}^*$. In this work, we focus on general *atomic norms* $R(\cdot)$. Given a set of atomic vectors $\mathcal{A} \subset \mathbb{R}^p$, the corresponding atomic norm of any $\boldsymbol{\theta} \in \mathbb{R}^p$ is given by

$$\|\boldsymbol{\theta}\|_{\mathcal{A}} = \inf\left\{\sum_{\mathbf{a}\in\mathcal{A}} c_{\mathbf{a}} \ : \ \boldsymbol{\theta} = \sum_{\mathbf{a}\in\mathcal{A}} c_{\mathbf{a}}\mathbf{a}, \ c_{\mathbf{a}} \geq 0 \ \forall \, \mathbf{a} \in \mathcal{A}\right\} \tag{6}$$

In order for $\|\cdot\|_{\mathcal{A}}$ to be a valid norm, atomic vectors in $\mathcal{A}$ has to span $\mathbb{R}^p$, and $\mathbf{a} \in \mathcal{A}$ iff $-\mathbf{a} \in \mathcal{A}$. The unit ball of atomic norm $\|\cdot\|_{\mathcal{A}}$ is given by $\Omega_{\mathcal{A}} = \text{conv}(\mathcal{A})$. In addition, we assume that the atomic set $\mathcal{A}$ contains $\mathbf{v} \odot \mathbf{a}$ for any $\mathbf{v} \in \{\pm 1\}^p$ if $\mathbf{a}$ belongs to $\mathcal{A}$, where $\odot$ denotes the elementwise (Hadamard) product for vectors. This assumption guarantees that both $\|\cdot\|_{\mathcal{A}}$ and its dual norm are *invariant under sign-changes*, which is satisfied by many widely used norms, such as $L_1$ norm, OWL norm [5] and $k$-support norm [2]. For the rest of the paper, we will use $\Omega_{\mathcal{A}}$, $\mathcal{T}_{\mathcal{A}}(\boldsymbol{\theta}^*)$, $\mathcal{C}_{\mathcal{A}}(\boldsymbol{\theta}^*)$ and $\Psi_{\mathcal{A}}(\boldsymbol{\theta}^*)$ with $\mathcal{A}$ replaced by appropriate subscript for specific norms. For any vector $\mathbf{u}$ and coordinate set $\mathcal{S}$, we define $\mathbf{u}_{\mathcal{S}}$ by zeroing out all the coordinates outside $\mathcal{S}$.

## 3 General Analysis for Atomic Norms

In this section, we present detailed analysis of the general bounds for the geometric measures, $w(\Omega_{\mathcal{A}})$, $w(\mathcal{C}_{\mathcal{A}}(\boldsymbol{\theta}^*))$ and $\Psi_{\mathcal{A}}(\boldsymbol{\theta}^*)$. In general, knowing the atomic set $\mathcal{A}$ is sufficient for bounding $w(\Omega_{\mathcal{A}})$. For $w(\mathcal{C}_{\mathcal{A}}(\boldsymbol{\theta}^*))$ and $\Psi_{\mathcal{A}}(\boldsymbol{\theta}^*)$, we only need a single subgradient of $\|\boldsymbol{\theta}^*\|_{\mathcal{A}}$ and some simple additional calculations.

### 3.1 Gaussian width of unit norm ball

Although the atomic set $\mathcal{A}$ may contain uncountably many vectors, we assume that $\mathcal{A}$ can be decomposed as a union of $M$ "simple" sets, $\mathcal{A} = \mathcal{A}_1 \cup \mathcal{A}_2 \cup \ldots \cup \mathcal{A}_M$. By "simple," we mean the Gaussian width of each $\mathcal{A}_i$ is easy to compute/bound. Such a decomposition assumption is often satisfied by commonly used atomic norms, e.g., $L_1$, $L_1/L_2$, OWL, $k$-support norm. The Gaussian width of the unit norm ball of $\|\cdot\|_{\mathcal{A}}$ can be easily obtained using the following lemma, which is essentially the Lemma 2 in [16]. Related results appear in [16].

**Lemma 2** *Let* $M > 4$, $\mathcal{A}_1, \cdots, \mathcal{A}_M \subset \mathbb{R}^p$, *and* $\mathcal{A} = \cup_m \mathcal{A}_m$. *The Gaussian width of unit norm ball of* $\|\cdot\|_{\mathcal{A}}$ *satisfies*

$$w(\Omega_{\mathcal{A}}) = w(conv(\mathcal{A})) = w(\mathcal{A}) \leq \max_{1 \leq m \leq M} w(\mathcal{A}_m) + 2\sup_{\mathbf{z}\in\mathcal{A}} \|\mathbf{z}\|_2 \sqrt{\log M} \tag{7}$$

Next we illustrate application of this result to bounding the width of the unit norm ball of $L_1$ and OWL norm.

**Example 1.1 ($L_1$ norm):** Recall that the $L_1$ norm can be viewed as the atomic norm induced by the set $\mathcal{A}_{L_1} = \{\pm\mathbf{e}_i \ : \ 1 \leq i \leq p\}$, where $\{\mathbf{e}_i\}_{i=1}^p$ is the canonical basis of $\mathbb{R}^p$. Since the Gaussian width of a singleton is 0, if we treat $\mathcal{A}$ as the union of individual $\{+\mathbf{e}_i\}$ and $\{-\mathbf{e}_i\}$, we have

$$w(\Omega_{L_1}) \leq 0 + 2\sqrt{\log 2p} = O(\sqrt{\log p}) . \tag{8}$$

**Example 1.2 (OWL norm):** A recent variant of $L_1$ norm is the so-called *ordered weighted $L_1$ (OWL)* norm [13, 25, 5] defined as $\|\boldsymbol{\theta}\|_{\text{owl}} = \sum_{i=1}^p w_i|\boldsymbol{\theta}|_i^{\downarrow}$, where $w_1 \geq w_2 \geq \ldots \geq w_p \geq 0$ are pre-specified ordered weights, and $|\boldsymbol{\theta}|^{\downarrow}$ is the permutation of $|\boldsymbol{\theta}|$ with entries sorted in decreasing order. In [25], the OWL norm is proved to be an atomic norm with atomic set

$$\mathcal{A}_{\text{owl}} = \bigcup_{1 \leq i \leq p} \mathcal{A}_i = \bigcup_{1 \leq i \leq p} \bigcup_{|\text{supp}(\mathcal{S})|=i} \left\{\mathbf{u} \in \mathbb{R}^p \ : \ \mathbf{u}_{\mathcal{S}^c} = \mathbf{0}, \mathbf{u}_{\mathcal{S}} = \frac{\mathbf{v}_{\mathcal{S}}}{\sum_{j=1}^i w_j}, \mathbf{v} \in \{\pm 1\}^p\right\} . \tag{9}$$

We first apply Lemma 2 to each set $\mathcal{A}_i$, and note that each $\mathcal{A}_i$ contains $2^i \binom{p}{i}$ atomic vectors.

$$w(\mathcal{A}_i) \le 0 + 2\sqrt{\frac{i}{(\sum_{j=1}^{i} w_j)^2}} \sqrt{\log 2^i \binom{p}{i}} \le \frac{2i}{\sum_{j=1}^{i} w_j}\sqrt{2 + \log\left(\frac{p}{i}\right)} \le \frac{2}{\bar{w}}\sqrt{2 + \log\left(\frac{p}{i}\right)},$$

where $\bar{w}$ is the average of $w_1, \ldots, w_p$. Then we apply the lemma again to $\mathcal{A}_{\text{owl}}$ and obtain

$$w(\Omega_{\text{owl}}) = w(\mathcal{A}_{\text{owl}}) \le \frac{2}{\bar{w}}\sqrt{2 + \log p} + \frac{2}{\bar{w}}\sqrt{\log p} = O\left(\frac{\sqrt{\log p}}{\bar{w}}\right), \tag{10}$$

which matches the result in [13].

## 3.2   Gaussian width of the intersection of tangent cone and unit sphere

In this subsection, we consider the computation of general $w(\mathcal{C}_{\mathcal{A}}(\boldsymbol{\theta}^*))$. Using the definition of dual norm, we can write $\|\boldsymbol{\theta}^*\|_{\mathcal{A}}$ as $\|\boldsymbol{\theta}^*\|_{\mathcal{A}} = \sup_{\|\mathbf{u}\|_{\mathcal{A}}^* \le 1} \langle \mathbf{u}, \boldsymbol{\theta}^* \rangle$, where $\|\cdot\|_{\mathcal{A}}^*$ denotes the dual norm of $\|\cdot\|_{\mathcal{A}}$. The $\mathbf{u}^*$ for which $\langle \mathbf{u}^*, \boldsymbol{\theta}^* \rangle = \|\boldsymbol{\theta}^*\|_{\mathcal{A}}$, is a subgradient of $\|\boldsymbol{\theta}^*\|_{\mathcal{A}}$. One can obtain $\mathbf{u}^*$ by simply solving the so-called *polar operator* [26] for the dual norm $\|\cdot\|_{\mathcal{A}}^*$,

$$\mathbf{u}^* \in \operatorname*{argmax}_{\|\mathbf{u}\|_{\mathcal{A}}^* \le 1} \langle \mathbf{u}, \boldsymbol{\theta}^* \rangle. \tag{11}$$

Based on polar operator, we start with the Lemma 3, which plays a key role in our analysis.

**Lemma 3** *Let $\mathbf{u}^*$ be a solution to the polar operator* (11)*, and define the weighted $L_1$ semi-norm $\|\cdot\|_{\mathbf{u}^*}$ as $\|\mathbf{v}\|_{\mathbf{u}^*} = \sum_{i=1}^{p} |u_i^*| \cdot |v_i|$. Then the following relation holds*

$$\mathcal{T}_{\mathcal{A}}(\boldsymbol{\theta}^*) \subseteq \mathcal{T}_{\mathbf{u}^*}(\boldsymbol{\theta}^*),$$

*where $\mathcal{T}_{\mathbf{u}^*}(\boldsymbol{\theta}^*) = \operatorname{cone}\{\mathbf{v} \in \mathbb{R}^p \mid \|\boldsymbol{\theta}^* + \mathbf{v}\|_{\mathbf{u}^*} \le \|\boldsymbol{\theta}^*\|_{\mathbf{u}^*}\}$.*

The proof of this lemma is in supplementary material. Note that the solution to (11) may not be unique. A good criterion for choosing $\mathbf{u}^*$ is to avoid zeros in $\mathbf{u}^*$, as any $u_i^* = 0$ will lead to the unboundedness of unit ball of $\|\cdot\|_{\mathbf{u}^*}$, which could potentially increase the size of $\mathcal{T}_{\mathbf{u}^*}(\boldsymbol{\theta}^*)$. Next we present the upper bound for $w(\mathcal{C}_{\mathcal{A}}(\boldsymbol{\theta}^*))$.

**Theorem 4** *Suppose that $\mathbf{u}^*$ is one of the solutions to* (11)*, and define the following sets,*

$$\mathcal{Q} = \{i \mid u_i^* = 0\}, \qquad \mathcal{S} = \{i \mid u_i^* \ne 0, \ \theta_i^* \ne 0\}, \qquad \mathcal{R} = \{i \mid u_i^* \ne 0, \ \theta_i^* = 0\}.$$

*The Gaussian width $w(\mathcal{C}_{\mathcal{A}}(\boldsymbol{\theta}^*))$ is upper bounded by*

$$w(\mathcal{C}_{\mathcal{A}}(\boldsymbol{\theta}^*)) \le \begin{cases} \sqrt{p}, & \text{if } \mathcal{R} \text{ is empty} \\ \sqrt{m + \frac{3}{2}s + \frac{2\kappa_{\max}^2}{\kappa_{\min}^2} s \log\left(\frac{p-m}{s}\right)}, & \text{if } \mathcal{R} \text{ is nonempty} \end{cases}, \tag{12}$$

*where $m = |\mathcal{Q}|$, $s = |\mathcal{S}|$, $\kappa_{\min} = \min_{i \in \mathcal{R}} |u_i^*|$ and $\kappa_{\max} = \max_{i \in \mathcal{S}} |u_i^*|$.*

*Proof:* By Lemma 3, we have $w(\mathcal{C}_{\mathcal{A}}(\boldsymbol{\theta}^*)) \le w(\mathcal{T}_{\mathbf{u}^*}(\boldsymbol{\theta}^*) \cap \mathbb{S}^{p-1}) \triangleq w(\mathcal{C}_{\mathbf{u}^*}(\boldsymbol{\theta}^*))$. Hence we can focus on bounding $w(\mathcal{C}_{\mathbf{u}^*}(\boldsymbol{\theta}^*))$. We first analyze the structure of $\mathbf{v}$ that satisfies $\|\boldsymbol{\theta}^* + \mathbf{v}\|_{\mathbf{u}^*} \le \|\boldsymbol{\theta}^*\|_{\mathbf{u}^*}$. For the coordinates $\mathcal{Q} = \{i \mid u_i^* = 0\}$, the corresponding entries $v_i$'s can be arbitrary since it does not affect the value of $\|\boldsymbol{\theta}^* + \mathbf{v}\|_{\mathbf{u}^*}$. Thus all possible $\mathbf{v}_{\mathcal{Q}}$ form a $m$-dimensional subspace, where $m = |\mathcal{Q}|$. For $\mathcal{S} \cup \mathcal{R} = \{i \mid u_i^* \ne 0\}$, we define $\tilde{\boldsymbol{\theta}} = \boldsymbol{\theta}_{\mathcal{S} \cup \mathcal{R}}^*$ and $\tilde{\mathbf{v}} = \mathbf{v}_{\mathcal{S} \cup \mathcal{R}}$, and $\tilde{\mathbf{v}}$ needs to satisfy

$$\|\tilde{\mathbf{v}} + \tilde{\boldsymbol{\theta}}\|_{\mathbf{u}^*} \le \|\tilde{\boldsymbol{\theta}}\|_{\mathbf{u}^*},$$

which is similar to the $L_1$-norm tangent cone except that coordinates are weighted by $|\mathbf{u}^*|$. Therefore we use the techniques for proving the Proposition 3.10 in [10]. Based on the structure of $\mathbf{v}$, The normal cone at $\boldsymbol{\theta}^*$ for $\mathcal{T}_{\mathbf{u}^*}(\boldsymbol{\theta}^*)$ is given by

$$\mathcal{N}(\boldsymbol{\theta}^*) = \{\mathbf{z} \ : \ \langle \mathbf{z}, \mathbf{v} \rangle \le 0 \ \forall \mathbf{v} \text{ s.t. } \|\mathbf{v} + \boldsymbol{\theta}^*\|_{\mathbf{u}^*} \le \|\boldsymbol{\theta}^*\|_{\mathbf{u}^*}\}$$

$$= \{\mathbf{z} \ : \ z_i = 0 \text{ for } i \in \mathcal{Q}, \ z_i = |u_i^*|\operatorname{sign}(\tilde{\theta}_i)t \text{ for } i \in \mathcal{S}, \ |z_i| \le |u_i^*|t \text{ for } i \in \mathcal{R}, \text{ for any } t \ge 0\}.$$

Given a standard Gaussian random vector $\mathbf{g}$, using the relation between Gaussian width and statistical dimension (Proposition 2.4 and 10.2 in [1]), we have

$$w^2(\mathcal{C}_{\mathbf{u}^*}(\boldsymbol{\theta}^*)) \le \mathbb{E}[\inf_{\mathbf{z}\in\mathcal{N}(\boldsymbol{\theta}^*)}\|\mathbf{z}-\mathbf{g}\|_2^2] = \mathbb{E}[\inf_{\mathbf{z}\in\mathcal{N}(\boldsymbol{\theta}^*)}\sum_{i\in\mathcal{Q}}g_i^2 + \sum_{j\in\mathcal{S}}(z_j-g_j)^2 + \sum_{k\in\mathcal{R}}(z_k-g_k)^2]$$

$$= |\mathcal{Q}| + \mathbb{E}[\inf_{\mathbf{z}_{\mathcal{S}\cup\mathcal{R}}\in\mathcal{N}(\boldsymbol{\theta}^*)}\sum_{j\in\mathcal{S}}(|u_j^*|\mathrm{sign}(\tilde{\theta}_j)t - g_j)^2 + \sum_{k\in\mathcal{R}}(z_k-g_k)^2]$$

$$\le |\mathcal{Q}| + t^2\sum_{j\in\mathcal{S}}|u_j^*|^2 + |\mathcal{S}| + \mathbb{E}[\sum_{k\in\mathcal{R}}\inf_{|z_k|\le|u_k^*|t}(z_k-g_k)^2]$$

$$\le |\mathcal{Q}| + t^2\sum_{j\in\mathcal{S}}|u_j^*|^2 + |\mathcal{S}| + \sum_{k\in\mathcal{R}}\frac{2}{\sqrt{2\pi}}\left(\int_{|u_k^*|t}^{+\infty}(g_k-|u_k^*|t)^2\exp(\frac{-g_k^2}{2})dg_k\right)$$

$$\le |\mathcal{Q}| + t^2\sum_{j\in\mathcal{S}}|u_j^*|^2 + |\mathcal{S}| + \sum_{k\in\mathcal{R}}\frac{2}{\sqrt{2\pi}}\frac{1}{|u_k^*|t}\exp\left(-\frac{|u_k^*|^2t^2}{2}\right) \quad (*) .$$

The details for the derivation above can be found in Appendix C of [10]. If $\mathcal{R}$ is empty, by taking $t = 0$, we have

$$(*) \le |\mathcal{Q}| + t^2\sum_{j\in\mathcal{S}}|u_j^*|^2 + |\mathcal{S}| = |\mathcal{Q}| + |\mathcal{S}| = p .$$

If $\mathcal{R}$ is nonempty, we denote $\kappa_{\min} = \min_{i\in\mathcal{R}}|u_i^*|$ and $\kappa_{\max} = \max_{i\in\mathcal{S}}|u_i^*|$. Taking $t = \frac{1}{\kappa_{\min}}\sqrt{2\log\left(\frac{|\mathcal{S}\cup\mathcal{R}|}{|\mathcal{S}|}\right)}$, we obtain

$$(*) \le |\mathcal{Q}| + |\mathcal{S}|(\kappa_{\max}^2t^2+1) + \frac{2|\mathcal{R}|\exp\left(-\frac{\kappa_{\min}^2t^2}{2}\right)}{\sqrt{2\pi}\kappa_{\min}t} = |\mathcal{Q}| + |\mathcal{S}|\left(\frac{2\kappa_{\max}^2}{\kappa_{\min}^2}\log\left(\frac{|\mathcal{S}\cup\mathcal{R}|}{|\mathcal{S}|}\right)+1\right)$$

$$+ \frac{|\mathcal{R}||\mathcal{S}|}{|\mathcal{S}\cup\mathcal{R}|\sqrt{\pi\log\left(\frac{|\mathcal{S}\cup\mathcal{R}|}{|\mathcal{S}|}\right)}} \le |\mathcal{Q}| + \frac{2\kappa_{\max}^2}{\kappa_{\min}^2}|\mathcal{S}|\log\left(\frac{|\mathcal{S}\cup\mathcal{R}|}{|\mathcal{S}|}\right) + \frac{3}{2}|\mathcal{S}| .$$

Substituting $|\mathcal{Q}| = m$, $|\mathcal{S}| = s$ and $|\mathcal{S}\cup\mathcal{R}| = p-m$ into the last inequality completes the proof. ∎

Suppose that $\boldsymbol{\theta}^*$ is a $s$-sparse vector. We illustrate the above bound on the Gaussian width of the spherical cap using $L_1$ norm and OWL norm as examples.

**Example 2.1 ($L_1$ norm):** The dual norm of $L_1$ is $L_\infty$ norm, and its easy to verify that $\mathbf{u}^* = [1,1,\ldots,1]^T \in \mathbb{R}^p$ is a solution to (11). Applying Theorem 4 to $\mathbf{u}^*$, we have

$$w(\mathcal{C}_{L_1}(\boldsymbol{\theta}^*)) \le \sqrt{\frac{3}{2}s + 2s\log\left(\frac{p}{s}\right)} = O\left(\sqrt{s + s\log\left(\frac{p}{s}\right)}\right) .$$

**Example 2.2 (OWL norm):** For OWL, its dual norm is given by $\|\mathbf{u}\|_{\mathrm{owl}}^* = \max_{\mathbf{b}\in\mathcal{A}_{\mathrm{owl}}}\langle\mathbf{b},\mathbf{u}\rangle$. W.l.o.g. we assume $\boldsymbol{\theta}^* = |\boldsymbol{\theta}^*|^{\downarrow}$, and a solution to (11) is given by $\mathbf{u}^* = [w_1,\ldots,w_s,\tilde{w},\tilde{w},\ldots,\tilde{w}]^T$, in which $\tilde{w}$ is the average of $w_{s+1},\ldots,w_p$. If all $w_i$'s are nonzero, the Gaussian width satisfies

$$w(\mathcal{C}_{\mathrm{owl}}(\boldsymbol{\theta}^*)) \le \sqrt{\frac{3}{2}s + \frac{2w_1^2}{\tilde{w}^2}s\log\left(\frac{p}{s}\right)} .$$

### 3.3 Restricted norm compatibility

The next theorem gives general upper bounds for the restricted norm compatibility $\Psi_{\mathcal{A}}(\boldsymbol{\theta}^*)$.

**Theorem 5** *Assume that $\|\mathbf{u}\|_{\mathcal{A}} \le \max\{\beta_1\|\mathbf{u}\|_1, \beta_2\|\mathbf{u}\|_2\}$ for all $\mathbf{u} \in \mathbb{R}^p$. Under the setting of Theorem 4, the restricted norm compatibility $\Psi_{\mathcal{A}}(\boldsymbol{\theta}^*)$ is upper bounded by*

$$\Psi_{\mathcal{A}}(\boldsymbol{\theta}^*) \le \begin{cases} \Phi , & \text{if } \mathcal{R} \text{ is empty} \\ \Phi_{\mathcal{Q}} + \max\left\{\beta_2, \beta_1\left(1+\frac{\kappa_{\max}}{\kappa_{\min}}\right)\sqrt{s}\right\} , & \text{if } \mathcal{R} \text{ is nonempty} \end{cases} , \qquad (13)$$

*where $\Phi = \sup_{\mathbf{u}\in\mathbb{R}^p}\frac{\|\mathbf{u}\|_{\mathcal{A}}}{\|\mathbf{u}\|_2}$ and $\Phi_{\mathcal{Q}} = \sup_{\mathrm{supp}(\mathbf{u})\subseteq\mathcal{Q}}\frac{\|\mathbf{u}\|_{\mathcal{A}}}{\|\mathbf{u}\|_2}$.*

*Proof:* As analyzed in the proof of Theorem 4, $\mathbf{v}_{\mathcal{Q}}$ for $\mathbf{v} \in \mathcal{T}_{\mathbf{u}^*}(\boldsymbol{\theta}^*)$ can be arbitrary, and the $\mathbf{v}_{\mathcal{S} \cup \mathcal{R}} = \mathbf{v}_{\mathcal{Q}^c}$ satisfies

$$\|\mathbf{v}_{\mathcal{Q}^c} + \boldsymbol{\theta}^*_{\mathcal{Q}^c}\|_{\mathbf{u}^*} \leq \|\boldsymbol{\theta}^*_{\mathcal{Q}^c}\|_{\mathbf{u}^*} \implies \sum_{i \in \mathcal{S}} |\theta^*_i + v_i||u^*_i| + \sum_{j \in \mathcal{R}} |v_j||u^*_j| \leq \sum_{i \in \mathcal{S}} |\theta^*_i||u^*_i|$$

$$\implies \sum_{i \in \mathcal{S}} (|\theta^*_i| - |v_i|)|u^*_i| + \sum_{j \in \mathcal{R}} |v_j||u^*_j| \leq \sum_{i \in \mathcal{S}} |\theta^*_i||u^*_i| \implies \kappa_{\min}\|\mathbf{v}_{\mathcal{R}}\|_1 \leq \kappa_{\max}\|\mathbf{v}_{\mathcal{S}}\|_1$$

If $\mathcal{R}$ is empty, by Lemma 3, we obtain

$$\Psi_{\mathcal{A}}(\boldsymbol{\theta}^*) \leq \Psi_{\mathbf{u}^*}(\boldsymbol{\theta}^*) \triangleq \sup_{\mathbf{v} \in \mathcal{T}_{\mathbf{u}^*}(\boldsymbol{\theta}^*)} \frac{\|\mathbf{v}\|_{\mathcal{A}}}{\|\mathbf{v}\|_2} \leq \sup_{\mathbf{v} \in \mathbb{R}^p} \frac{\|\mathbf{v}\|_{\mathcal{A}}}{\|\mathbf{v}\|_2} = \Phi .$$

If $\mathcal{R}$ is nonempty, we have

$$\Psi_{\mathcal{A}}(\boldsymbol{\theta}^*) \leq \Psi_{\mathbf{u}^*}(\boldsymbol{\theta}^*) \leq \sup_{\mathbf{v} \in \mathcal{T}_{\mathbf{u}^*}(\boldsymbol{\theta}^*)} \frac{\|\mathbf{v}_{\mathcal{Q}}\|_{\mathcal{A}} + \|\mathbf{v}_{\mathcal{Q}^c}\|_{\mathcal{A}}}{\|\mathbf{v}\|_2} \leq \sup_{\substack{\text{supp}(\mathbf{v}) \subseteq \mathcal{Q}, \, \text{supp}(\mathbf{v}') \subseteq \mathcal{Q}^c \\ \kappa_{\min}\|\mathbf{v}'_{\mathcal{R}}\|_1 \leq \kappa_{\max}\|\mathbf{v}'_{\mathcal{S}}\|_1}} \frac{\|\mathbf{v}\|_{\mathcal{A}} + \|\mathbf{v}'\|_{\mathcal{A}}}{\|\mathbf{v} + \mathbf{v}'\|_2}$$

$$\leq \sup_{\substack{\text{supp}(\mathbf{v}) \subseteq \mathcal{Q}}} \frac{\|\mathbf{v}\|_{\mathcal{A}}}{\|\mathbf{v}\|_2} + \sup_{\substack{\text{supp}(\mathbf{v}') \subseteq \mathcal{Q}^c \\ \kappa_{\min}\|\mathbf{v}'_{\mathcal{R}}\|_1 \leq \kappa_{\max}\|\mathbf{v}'_{\mathcal{S}}\|_1}} \frac{\max\{\beta_1\|\mathbf{v}'\|_1, \beta_2\|\mathbf{v}'\|_2\}}{\|\mathbf{v}'\|_2}$$

$$\leq \Phi_{\mathcal{Q}} + \max\{\beta_2, \sup_{\text{supp}(\mathbf{v}') \subseteq \mathcal{S}} \frac{\beta(1 + \frac{\kappa_{\max}}{\kappa_{\min}})\|\mathbf{v}'\|_1}{\|\mathbf{v}'\|_2}\} \leq \Phi_{\mathcal{Q}} + \max\{\beta_2, \beta_1\left(1 + \frac{\kappa_{\max}}{\kappa_{\min}}\right)\sqrt{s}\} ,$$

in which the last inequality in the first line uses the property of $\mathcal{T}_{\mathbf{u}^*}(\boldsymbol{\theta}^*)$. ∎

**Remark:** We call $\Phi$ the *unrestricted norm compatibility*, and $\Phi_{\mathcal{Q}}$ the *subspace norm compatibility*, both of which are often easier to compute than $\Psi_{\mathcal{A}}(\boldsymbol{\theta}^*)$. The $\beta_1$ and $\beta_2$ in the assumption of $\|\cdot\|_{\mathcal{A}}$ can have multiple choices, and one has the flexibility to choose the one that yields the tightest bound.

**Example 3.1 ($L_1$ norm):** To apply the Theorem 5 to $L_1$ norm, we can choose $\beta_1 = 1$ and $\beta_2 = 0$. We recall the $\mathbf{u}^*$ for $L_1$ norm, whose $\mathcal{Q}$ is empty while $\mathcal{R}$ is nonempty. So we have for $s$-sparse $\boldsymbol{\theta}^*$

$$\Psi_{L_1}(\boldsymbol{\theta}^*) \leq 0 + \max\left\{0, \left(1 + \frac{1}{1}\right)\sqrt{s}\right\} = 2\sqrt{s} .$$

**Example 3.2 (OWL norm):** For OWL, note that $\|\cdot\|_{\text{owl}} \leq w_1\|\cdot\|_1$. Hence we choose $\beta_1 = w_1$ and $\beta_2 = 0$. As a result, we similarly have for $s$-sparse $\boldsymbol{\theta}^*$

$$\Psi_{\text{owl}}(\boldsymbol{\theta}^*) \leq 0 + \max\left\{0, w_1\left(1 + \frac{w_1}{\tilde{w}}\right)\sqrt{s}\right\} \leq \frac{2w_1^2}{\tilde{w}}\sqrt{s} .$$

## 4 Tightness of the General Bounds

So far we have shown that the geometric measures can be upper bounded for general atomic norms. One might wonder how tight the bounds in Section 3 are for these measures. For $w(\Omega_{\mathcal{A}})$, as the result from [16] depends on the decomposition of $\mathcal{A}$ for the ease of computation, it might be tricky to discuss its tightness in general. Hence we will focus on the other two, $w(\mathcal{C}_{\mathcal{A}}(\boldsymbol{\theta}^*))$ and $\Psi_{\mathcal{A}}(\boldsymbol{\theta}^*)$.

To characterize the tightness, we need to compare the lower bounds of $w(\mathcal{C}_{\mathcal{A}}(\boldsymbol{\theta}^*))$ and $\Psi_{\mathcal{A}}(\boldsymbol{\theta}^*)$, with their upper bounds determined by $\mathbf{u}^*$. While there can be multiple $\mathbf{u}^*$, it is easy to see that any convex combination of them is also a solution to (11). Therefore we can always find a $\mathbf{u}^*$ that has the largest support, i.e., $\text{supp}(\mathbf{u}') \subseteq \text{supp}(\mathbf{u}^*)$ for any other solution $\mathbf{u}'$. We will use such $\mathbf{u}^*$ to generate the lower bounds. First we need the following lemma for the cone $\mathcal{T}_{\mathcal{A}}(\boldsymbol{\theta}^*)$.

**Lemma 6** *Consider a solution $\mathbf{u}^*$ to (11), which satisfies $\text{supp}(\mathbf{u}') \subseteq \text{supp}(\mathbf{u}^*)$ for any other solution $\mathbf{u}'$. Under the setting of notations in Theorem 4, we define an additional set of coordinates $\mathcal{P} = \{i \mid u^*_i = 0, \theta^*_i = 0\}$. Then the tangent cone $\mathcal{T}_{\mathcal{A}}(\boldsymbol{\theta}^*)$ satisfies*

$$\mathcal{T}_1 \oplus \mathcal{T}_2 \subseteq \text{cl}(\mathcal{T}_{\mathcal{A}}(\boldsymbol{\theta}^*)) , \tag{14}$$

*where $\oplus$ denotes the direct (Minkowski) sum operation, $\text{cl}(\cdot)$ denotes the closure, $\mathcal{T}_1 = \{\mathbf{v} \in \mathbb{R}^p \mid v_i = 0 \text{ for } i \notin \mathcal{P}\}$ is a $|\mathcal{P}|$-dimensional subspace, and $\mathcal{T}_2 = \{\mathbf{v} \in \mathbb{R}^p \mid \text{sign}(v_i) = -\text{sign}(\theta^*_i) \text{ for } i \in \text{supp}(\boldsymbol{\theta}^*), v_i = 0 \text{ for } i \notin \text{supp}(\boldsymbol{\theta}^*)\}$ is a $|\text{supp}(\boldsymbol{\theta}^*)|$-dimensional orthant.*

The proof of Lemma 6 is given in supplementary material. The following theorem gives us the lower bound for $w(\mathcal{C}_{\mathcal{A}}(\boldsymbol{\theta}^*))$ and $\Psi_{\mathcal{A}}(\boldsymbol{\theta}^*)$.

**Theorem 7** *Under the setting of Theorem 4 and Lemma 6, the following lower bounds hold,*

$$w(\mathcal{C}_{\mathcal{A}}(\boldsymbol{\theta}^*)) \geq O(\sqrt{m+s}) , \quad (15)$$

$$\Psi_{\mathcal{A}}(\boldsymbol{\theta}^*) \geq \Phi_{\mathcal{Q} \cup \mathcal{S}} . \quad (16)$$

*Proof:*  To lower bound $w(\mathcal{C}_{\mathcal{A}}(\boldsymbol{\theta}^*))$, we use Lemma 6 and the relation between Gaussian width and statistical dimension (Proposition 10.2 in [1]),

$$w(\mathcal{T}_{\mathcal{A}}(\boldsymbol{\theta}^*)) \geq w(\mathcal{T}_1 \oplus \mathcal{T}_2 \cap \mathbb{S}^{p-1}) \geq \sqrt{\mathbb{E}[\inf_{\mathbf{z} \in \mathcal{N}_{\mathcal{T}_1 \oplus \mathcal{T}_2}(\boldsymbol{\theta}^*)} \|\mathbf{z} - \mathbf{g}\|_2^2] - 1} \quad (*) ,$$

where the normal cone $\mathcal{N}_{\mathcal{T}_1 \oplus \mathcal{T}_2}(\boldsymbol{\theta}^*)$ of $\mathcal{T}_1 \oplus \mathcal{T}_2$ is given by $\mathcal{N}_{\mathcal{T}_1 \oplus \mathcal{T}_2}(\boldsymbol{\theta}^*) = \{\mathbf{z} \; : \; z_i = 0 \; \text{for} \; i \in \mathcal{P}, \; \text{sign}(z_i) = \text{sign}(\theta_i^*) \; \text{for} \; i \in \text{supp}(\boldsymbol{\theta}^*)\}$. Hence we have

$$(*) = \sqrt{\mathbb{E}[\sum_{i \in \mathcal{P}} g_i^2 + \sum_{j \in \text{supp}(\boldsymbol{\theta}^*)} g_j^2 \mathbb{I}_{\{g_j \theta_j^* < 0\}}] - 1} = \sqrt{|\mathcal{P}| + \frac{|\text{supp}(\boldsymbol{\theta}^*)|}{2} - 1} = O(\sqrt{m+s}) ,$$

where the last equality follows the fact that $\mathcal{P} \cup \text{supp}(\boldsymbol{\theta}^*) = \mathcal{Q} \cup \mathcal{S}$. This completes proof of (15). To prove (16), we again use Lemma 6 and the fact $\mathcal{P} \cup \text{supp}(\boldsymbol{\theta}^*) = \mathcal{Q} \cup \mathcal{S}$. Noting that $\|\cdot\|_{\mathcal{A}}$ is invariant under sign-changes, we get

$$\Psi_{\mathcal{A}}(\boldsymbol{\theta}^*) = \sup_{\mathbf{v} \in \mathcal{T}_{\mathcal{A}}(\boldsymbol{\theta}^*)} \frac{\|\mathbf{v}\|_{\mathcal{A}}}{\|\mathbf{v}\|_2} \geq \sup_{\mathbf{v} \in \mathcal{T}_1 \oplus \mathcal{T}_2} \frac{\|\mathbf{v}\|_{\mathcal{A}}}{\|\mathbf{v}\|_2} = \sup_{\text{supp}(\mathbf{v}) \subseteq \mathcal{P} \cup \text{supp}(\boldsymbol{\theta}^*)} \frac{\|\mathbf{v}\|_{\mathcal{A}}}{\|\mathbf{v}\|_2} = \Phi_{\mathcal{Q} \cup \mathcal{S}} . \qquad \blacksquare$$

**Remark:** We compare the lower bounds (15) (16) with the upper bounds (12) (13). If $\mathcal{R}$ is empty, $m + s = p$, and the lower bounds actually match the upper bounds up to a constant factor for both $w(\mathcal{C}_{\mathcal{A}}(\boldsymbol{\theta}^*))$ and $\Psi_{\mathcal{A}}(\boldsymbol{\theta}^*)$. If $\mathcal{R}$ is nonempty, the lower and upper bounds of $w(\mathcal{C}_{\mathcal{A}}(\boldsymbol{\theta}^*))$ differ by a multiplicative factor $\frac{2\kappa_{\max}^2}{\kappa_{\min}^2} \log(\frac{p-m}{s})$, which can be small in practice. For $\Psi_{\mathcal{A}}(\boldsymbol{\theta}^*)$, as $\Phi_{\mathcal{Q} \cup \mathcal{S}} \geq \Phi_{\mathcal{Q}}$, we usually have at most an additive $O(\sqrt{s})$ term in upper bound, since the assumption on $\|\cdot\|_{\mathcal{A}}$ often holds with a constant $\beta_1$ and $\beta_2 = 0$ for most norms.

## 5  Application to the $k$-Support Norm

In this section, we apply our general results on geometric measures to a non-trivial example, $k$-support norm [2], which has been proved effective for sparse recovery [11, 17, 12]. The $k$-support norm can be viewed as an atomic norm, for which $\mathcal{A} = \{\mathbf{a} \in \mathbb{R}^p \mid \|\mathbf{a}\|_0 \leq k, \; \|\mathbf{a}\|_2 \leq 1\}$. The $k$-support norm can be explicitly expressed as an infimum convolution given by

$$\|\boldsymbol{\theta}\|_k^{sp} = \inf_{\sum_i \mathbf{u}_i = \boldsymbol{\theta}} \left\{ \sum_i \|\mathbf{u}_i\|_2 \; \middle| \; \|\mathbf{u}_i\|_0 \leq k \right\} , \quad (17)$$

and its dual norm is the so-called 2-$k$ *symmetric gauge norm* defined as

$$\|\boldsymbol{\theta}\|_k^{sp^*} = \|\boldsymbol{\theta}\|_{(k)} = \||\boldsymbol{\theta}|_{1:k}^{\downarrow}\|_2 , \quad (18)$$

It is straightforward to see that the dual norm is simply the $L_2$ norm of the largest $k$ entries in $|\boldsymbol{\theta}|$. Suppose that all the sets of coordinates with cardinality $k$ can be listed as $\mathcal{S}_1, \mathcal{S}_2, \ldots, \mathcal{S}_{\binom{p}{k}}$. Then $\mathcal{A}$ can be written as $\mathcal{A} = \mathcal{A}_1 \cup \ldots \cup \mathcal{A}_{\binom{p}{k}}$, where each $\mathcal{A}_i = \{\mathbf{a} \in \mathbb{R}^p \mid \text{supp}(\mathbf{a}) \subseteq \mathcal{S}_i, \; \|\mathbf{a}\|_2 \leq 1\}$. It is not difficult to see that $w(\mathcal{A}_i) = \mathbb{E}[\sup_{\mathbf{a} \in \mathcal{A}_i} \langle \mathbf{a}, \mathbf{g} \rangle] = \mathbb{E}\|\mathbf{g}_{\mathcal{S}_i}\|_2 \leq \sqrt{\mathbb{E}\|\mathbf{g}_{\mathcal{S}_i}\|_2^2} \leq \sqrt{k}$. Using Lemma 2, we know the Gaussian width of the unit ball of $k$-support norm

$$w(\Omega_k^{sp}) \leq \sqrt{k} + 2\sqrt{\log\binom{p}{k}} \leq \sqrt{k} + 2\sqrt{k \log\left(\frac{p}{k}\right) + k} = O\left(\sqrt{k \log\left(\frac{p}{k}\right) + k}\right) , \quad (19)$$

which matches that in [11]. Now we turn to the calculation of $w(\mathcal{C}_k^{sp}(\boldsymbol{\theta}^*))$ and $\Psi_k^{sp}(\boldsymbol{\theta}^*)$. As we have seen in the general analysis, the solution $\mathbf{u}^*$ to the polar operator (11) is important in characterizing the two quantities. We first present a simple procedure in Algorithm 1 for solving the polar operator for $\|\cdot\|_k^{sp^*}$. The time complexity is only $O(p \log p + k)$. This procedure can be utilized to compute the $k$-support norm, or be applied to estimation with $\|\cdot\|_k^{sp^*}$ using *generalized conditional gradient* method [26], which requires solving the polar operator in each iteration.

---

**Algorithm 1** Solving polar operator for $\| \cdot \|_k^{sp^*}$

---

**Input:** $\boldsymbol{\theta}^* \in \mathbb{R}^p$, positive integer $k$
**Output:** solution $\mathbf{u}^*$ to the polar operator (11)
  1: $\mathbf{z} = |\boldsymbol{\theta}^*|^{\downarrow}$, $t = 0$
  2: **for** $i = 1$ to $k$ **do**
  3: $\quad \gamma_1 = \|\mathbf{z}_{1:i-1}\|_2$, $\gamma_2 = \|\mathbf{z}_{i:p}\|_1$, $d = k - i + 1$, $\beta = \frac{\gamma_2}{\sqrt{\gamma_2^2 d + \gamma_1^2 d^2}}$, $\alpha = \frac{\gamma_1}{2\sqrt{1 - \beta^2 d}}$, $\mathbf{w} = \frac{\mathbf{z}_{1:i-1}}{2\alpha}$
  4: $\quad$ **if** $\frac{\gamma_1^2}{2\alpha} + \beta\gamma_2 > t$ and $\beta < w_{i-1}$ **then**
  5: $\quad\quad t = \frac{\gamma_1^2}{2\alpha} + \beta\gamma_2$, $\mathbf{u}^* = [\mathbf{w}, \beta\mathbf{1}]^T$ $\quad$ ($\mathbf{1}$ is $(p - i + 1)$-dimensional vector with all ones)
  6: $\quad$ **end if**
  7: **end for**
  8: change the sign and order of $\mathbf{u}^*$ to conform with $\boldsymbol{\theta}^*$
  9: **return** $\mathbf{u}^*$

---

**Theorem 8** *For a given $\boldsymbol{\theta}^*$, Algorithm 1 returns a solution to polar operator* (11) *for* $\| \cdot \|_k^{sp^*}$.

The proof of this theorem is provided in supplementary material. Now we consider $w(\mathcal{C}_k^{sp}(\boldsymbol{\theta}^*))$ and $\Psi_k^{sp}(\boldsymbol{\theta}^*)$ for $s$-sparse $\boldsymbol{\theta}^*$ (here $s$-sparse $\boldsymbol{\theta}^*$ means $|\text{supp}(\boldsymbol{\theta}^*)| = s$) in three scenarios: (i) over-specified $k$, where $s < k$, (ii) exactly specified $k$, where $s = k$, and (iii) under-specified $k$, where $s > k$. The bounds are given in Theorem 9, and the proof is also in supplementary material.

**Theorem 9** *For given $s$-sparse $\boldsymbol{\theta}^* \in \mathbb{R}^p$, the Gaussian width $w(\mathcal{C}_k^{sp}(\boldsymbol{\theta}^*))$ and the restricted norm compatibility $\Psi_k^{sp}(\boldsymbol{\theta}^*)$ for a specified $k$ are given by*

$$
w(\mathcal{C}_k^{sp}(\boldsymbol{\theta}^*)) \leq
\begin{cases}
\sqrt{p} \,, & \text{if } s < k \\[2mm]
\sqrt{\frac{3}{2}s + \frac{2\theta_{\max}^{*2}}{\theta_{\min}^{*2}}s \log\left(\frac{p}{s}\right)} \,, & \text{if } s = k \\[2mm]
\sqrt{\frac{3}{2}s + \frac{2\kappa_{\max}^2}{\kappa_{\min}^2}s \log\left(\frac{p}{s}\right)} \,, & \text{if } s > k
\end{cases}
\,,\quad
\Psi_k^{sp}(\boldsymbol{\theta}^*) \leq
\begin{cases}
\sqrt{\frac{2p}{k}} \,, & \text{if } s < k \\[2mm]
\sqrt{2}(1 + \frac{\theta_{\max}^*}{\theta_{\min}^*}) \,, & \text{if } s = k \\[2mm]
(1 + \frac{\kappa_{\max}}{\kappa_{\min}})\sqrt{\frac{2s}{k}} \,, & \text{if } s > k
\end{cases}
\,,
$$

(20)

*where $\theta_{\max}^* = \max_{i \in \text{supp}(\boldsymbol{\theta}^*)} |\theta_i^*|$ and $\theta_{\min}^* = \min_{i \in \text{supp}(\boldsymbol{\theta}^*)} |\theta_i^*|$.*

**Remark:** Previously $\Psi_k^{sp}(\boldsymbol{\theta}^*)$ is unknown and the bound on $w(\mathcal{C}_k^{sp}(\boldsymbol{\theta}^*))$ given in [11] is loose, as it used the result in [21]. Based on Theorem 9, we note that the choice of $k$ can affect the recovery guarantees. Over-specified $k$ leads to a direct dependence on the dimensionality $p$ for $w(\mathcal{C}_k^{sp}(\boldsymbol{\theta}^*))$ and $\Psi_k^{sp}(\boldsymbol{\theta}^*)$, resulting in a weak error bound. The bounds are sharp for exactly specified or under-specified $k$. Thus, it is better to under-specify $k$ in practice. where the estimation error satifies

$$
\|\hat{\boldsymbol{\theta}}_{\lambda_n} - \boldsymbol{\theta}^*\|_2 \leq O\left(\sqrt{\frac{s + s \log(p/k)}{n}}\right)
$$

(21)

## 6 Conclusions

In this work, we study the problem of structured estimation with general atomic norms that are invariant under sign-changes. Based on Dantzig-type estimators, we provide the general bounds for the geometric measures. In terms of $w(\Omega_{\mathcal{A}})$, instead of comparison with other results or direct calculation, we demonstrate a third way to compute it based on decomposition of atomic set $\mathcal{A}$. For $w(\mathcal{C}_{\mathcal{A}}(\boldsymbol{\theta}^*))$ and $\Psi_{\mathcal{A}}(\boldsymbol{\theta}^*)$, we derive general upper bounds, which only require the knowledge of a single subgradient of $\|\boldsymbol{\theta}^*\|_{\mathcal{A}}$. We also show that these upper bounds are close to the lower bounds, which makes them practical in general. To illustrate our results, we discuss the application to $k$-support norm in details and shed light on the choice of $k$ in practice.

## Acknowledgements

The research was supported by NSF grants IIS-1447566, IIS-1422557, CCF-1451986, CNS-1314560, IIS-0953274, IIS-1029711, and by NASA grant NNX12AQ39A.

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
