[Supplementary Material]

# Supplementary Material to Structured Estimation with Atomic Norms: General Bounds and Applications

**Sheng Chen**        **Arindam Banerjee**
Dept. of Computer Science & Engg., University of Minnesota, Twin Cities
{shengc,banerjee}@cs.umn.edu

## 1 Proof of Lemma 3

**Statement of Lemma:** *Let $\mathbf{u}^*$ be a solution to the polar operator* (11)*, and define the weighted $L_1$ semi-norm $\| \cdot \|_{\mathbf{u}^*}$ as $\|\mathbf{v}\|_{\mathbf{u}^*} = \sum_{i=1}^{p} |u_i^*| \cdot |v_i|$. Then the following relation holds*

$$\mathcal{T}_{\mathcal{A}}(\boldsymbol{\theta}^*) \subseteq \mathcal{T}_{\mathbf{u}^*}(\boldsymbol{\theta}^*) \,,$$

*where $\mathcal{T}_{\mathbf{u}^*}(\boldsymbol{\theta}^*) = \text{cone}\{\mathbf{v} \in \mathbb{R}^p \mid \|\boldsymbol{\theta}^* + \mathbf{v}\|_{\mathbf{u}^*} \leq \|\boldsymbol{\theta}^*\|_{\mathbf{u}^*}\}$.*

*Proof:* As both $\mathcal{T}_{\mathcal{A}}(\boldsymbol{\theta}^*)$ and $\mathcal{T}_{\mathbf{u}^*}(\boldsymbol{\theta}^*)$ are cones, it is sufficient to show that $\{\mathbf{v} \mid \|\mathbf{v}\|_{\mathcal{A}} \leq \|\boldsymbol{\theta}^*\|_{\mathcal{A}}\} \subseteq \{\mathbf{v} \mid \|\mathbf{v}\|_{\mathbf{u}^*} \leq \|\boldsymbol{\theta}^*\|_{\mathbf{u}^*}\}$. Since $\|\boldsymbol{\theta}^*\|_{\mathbf{u}^*} = \|\boldsymbol{\theta}^*\|_{\mathcal{A}}$, it also suffices to show that $\{\mathbf{v} \mid \|\mathbf{v}\|_{\mathcal{A}} \leq 1\} \subseteq \{\mathbf{v} \mid \|\mathbf{v}\|_{\mathbf{u}^*} \leq 1\}$, i.e., the $\|\mathbf{v}\|_{\mathcal{A}} \geq \|\mathbf{v}\|_{\mathbf{u}^*}$ for $\mathbf{v} \in \mathbb{R}^p$. Using the dual norm definition and sign-change invariance of $\| \cdot \|_{\mathcal{A}}^*$, we obtain

$$\|\mathbf{v}\|_{\mathcal{A}} = \sup_{\|\mathbf{a}\|_{\mathcal{A}}^* \leq 1} \langle \mathbf{a}, \mathbf{v} \rangle \geq \langle \text{sign}(\mathbf{v}) \odot |\mathbf{u}^*|, \mathbf{v} \rangle = \langle |\mathbf{u}^*|, |\mathbf{v}| \rangle = \|\mathbf{v}\|_{\mathbf{u}^*} \,,$$

thus $\mathcal{T}_{\mathcal{A}}(\boldsymbol{\theta}^*) \subseteq \mathcal{T}_{\mathbf{u}^*}(\boldsymbol{\theta}^*)$. ∎

## 2 Proof of Lemma 6

**Statement of Lemma:** *Consider a solution $\mathbf{u}^*$ to* (11)*, which satisfies $\text{supp}(\mathbf{u}') \subseteq \text{supp}(\mathbf{u}^*)$ for any other solution $\mathbf{u}'$. Under the setting of notations in Theorem 4, we define an additional set of coordinates $\mathcal{P} = \{i \mid u_i^* = 0, \ \theta_i^* = 0\}$. Then the tangent cone $\mathcal{T}_{\mathcal{A}}(\boldsymbol{\theta}^*)$ satisfies*

$$\mathcal{T}_1 \oplus \mathcal{T}_2 \subseteq \text{cl}(\mathcal{T}_{\mathcal{A}}(\boldsymbol{\theta}^*)) \,, \tag{S.1}$$

*where $\oplus$ denotes the direct (Minkowski) sum operation, $\text{cl}(\cdot)$ denotes the closure, $\mathcal{T}_1 = \{\mathbf{v} \in \mathbb{R}^p \mid v_i = 0 \text{ for } i \notin \mathcal{P}\}$ is a $|\mathcal{P}|$-dimensional subspace, and $\mathcal{T}_2 = \{\mathbf{v} \in \mathbb{R}^p \mid \text{sign}(v_i) = -\text{sign}(\theta_i^*) \text{ for } i \in \text{supp}(\boldsymbol{\theta}^*), \ v_i = 0 \text{ for } i \notin \text{supp}(\boldsymbol{\theta}^*)\}$ is a $|\text{supp}(\boldsymbol{\theta}^*)|$-dimensional orthant.*

*Proof:* For any fixed $\boldsymbol{\theta}^* \in \mathbb{R}^p$ and its $\mathcal{P}$, we define a vector sequence $\{\mathbf{v}^{(k)} = \delta^{(k)}\mathbf{w}\}$ based on a given $\mathbf{w} \in \mathbb{R}^p$ and a monotonically decreasing positive scalar sequence $\{\delta^{(k)}\}$ with $\delta^{(1)} < \min_{i \in \text{supp}(\boldsymbol{\theta}^*)} |\theta_i^*|$ and $\lim_{k \to +\infty} \delta^{(k)} = 0$. $\mathbf{w}$ satisfies

$$w_i = \left\{ \begin{array}{ll} 0 \,, & \text{if } i \notin \mathcal{P} \cup \text{supp}(\boldsymbol{\theta}^*) \\ -\text{sign}(\theta_i^*) \,, & \text{if } i \in \text{supp}(\boldsymbol{\theta}^*) \\ \text{arbitrary} \,, & \text{if } i \in \mathcal{P} \end{array} \right. .$$

Let $\mathbf{u}^{(k)}$ be one solution to the polar operator for $\boldsymbol{\theta}^* + \mathbf{v}^{(k)}$, and form another sequence $\{\mathbf{u}^{(k)}\}$. Note that $\text{sign}(\theta_i^* + v_i^{(k)}) = \text{sign}(\theta_i^* - \text{sign}(\theta_i^*)\delta^{(k)}) = \text{sign}(\theta_i^*) = \text{sign}(u_i^{(k)})$ for $i \in \text{supp}(\boldsymbol{\theta}^*)$.

Then we have

$$\|\boldsymbol{\theta}^* + \mathbf{v}^{(k)}\|_{\mathcal{A}} - \|\boldsymbol{\theta}^*\|_{\mathcal{A}} \leq \langle \boldsymbol{\theta}^* + \mathbf{v}^{(k)}, \mathbf{u}^{(k)} \rangle - \langle \boldsymbol{\theta}^*, \mathbf{u}^{(k)} \rangle = \langle \mathbf{v}^{(k)}, \mathbf{u}^{(k)} \rangle$$

$$\leq -\delta^{(k)}\|\mathbf{u}^{(k)}_{\text{supp}(\boldsymbol{\theta}^*)}\|_1 + \delta^{(k)}\langle \mathbf{w}_{\mathcal{P}}, \mathbf{u}^{(k)}_{\mathcal{P}} \rangle \leq -\delta^{(k)}(\|\mathbf{u}^{(k)}_{\text{supp}(\boldsymbol{\theta}^*)}\|_1 - \|\mathbf{w}_{\mathcal{P}}\|_\infty \|\mathbf{u}^{(k)}_{\mathcal{P}}\|_1)$$

As $\delta^{(k)}$ approaches 0, $\boldsymbol{\theta}^* + \mathbf{v}^{(k)}$ converges to $\boldsymbol{\theta}^*$, and a subsequence $\{\mathbf{u}^{(k_i)}\}$ of $\{\mathbf{u}^{(k)}\}$ will converge to a solution $\mathbf{u}'$ to the polar operator for $\boldsymbol{\theta}^*$. Hence $\lim_{i \to +\infty} \|\mathbf{u}^{(k_i)}_{\text{supp}(\boldsymbol{\theta}^*)}\|_1 = \|\mathbf{u}'_{\text{supp}(\boldsymbol{\theta}^*)}\|_1 > 0$, $\lim_{i \to +\infty} \|\mathbf{u}^{(k_i)}_{\mathcal{P}}\|_1 = \|\mathbf{u}'_{\mathcal{P}}\|_1 = 0$, and for large enough $k_i$, we have

$$\|\boldsymbol{\theta}^* + \mathbf{v}^{(k_i)}\|_{\mathcal{A}} - \|\boldsymbol{\theta}^*\|_{\mathcal{A}} \leq -\delta^{(k_i)}(\|\mathbf{u}^{(k_i)}_{\text{supp}(\boldsymbol{\theta}^*)}\|_1 - \|\mathbf{w}_{\mathcal{P}}\|_\infty \|\mathbf{u}^{(k_i)}_{\mathcal{P}}\|_1) \leq 0,$$

thus $\mathbf{v}^{(k_i)}$ belongs to $\mathcal{T}_{\mathcal{A}}(\boldsymbol{\theta}^*)$. Since $\mathbf{v}^{(k)} = \delta^{(k)}\mathbf{w}$, $\mathbf{w}$ also belongs to $\mathcal{T}_{\mathcal{A}}(\boldsymbol{\theta}^*)$.
Now we show $\mathcal{T}_1 \subseteq \text{cl}(\mathcal{T}_{\mathcal{A}}(\boldsymbol{\theta}^*))$. For any $\mathbf{a} \in \mathcal{T}_1 = \{\mathbf{v} \in \mathbb{R}^p \mid v_i = 0 \text{ for } i \notin \mathcal{P}\}$ and arbitrarily small $\xi > 0$, we construct $\mathbf{w}$ such that $w_i = \frac{a_i}{\xi}$ for $i \in \mathcal{P}$. Based on the argument above, this $\mathbf{w}$ is in $\mathcal{T}_{\mathcal{A}}(\boldsymbol{\theta}^*)$. Therefore $\mathbf{a}' \triangleq \xi \mathbf{w} \in \mathcal{T}_{\mathcal{A}}(\boldsymbol{\theta}^*)$, and $\|\mathbf{a} - \mathbf{a}'\|_2 \leq \sqrt{|\text{supp}(\boldsymbol{\theta}^*)|}\xi$, which can be arbitrarily close to 0. Therefore taking the closure of $\mathcal{T}_{\mathcal{A}}(\boldsymbol{\theta}^*)$ gives us $\mathcal{T}_1 \subseteq \text{cl}(\mathcal{T}_{\mathcal{A}}(\boldsymbol{\theta}^*))$.
Next we show $\mathcal{T}_2 \subseteq \mathcal{T}_{\mathcal{A}}(\boldsymbol{\theta}^*)$. For any coordinate $i \in \text{supp}(\boldsymbol{\theta}^*)$, construct $\mathbf{v} \in \mathbb{R}^p$ such that $v_i = -\theta_i^*$ and $v_j = 0$ for $j \neq i$, and $\boldsymbol{\theta}' \in \mathbb{R}^p$ such that $\theta_i' = -\theta_i^*$ and $\theta_j' = \theta_j^*$ for $j \neq i$. As the norm $\|\cdot\|_{\mathcal{A}}$ is invariant under sign-changes, we can verify that

$$\|\boldsymbol{\theta}^* + \mathbf{v}\|_{\mathcal{A}} = \|\frac{\boldsymbol{\theta}^* + \boldsymbol{\theta}'}{2}\| \leq \frac{1}{2}\|\boldsymbol{\theta}^*\|_{\mathcal{A}} + \frac{1}{2}\|\boldsymbol{\theta}'\|_{\mathcal{A}} = \|\boldsymbol{\theta}^*\|_{\mathcal{A}} .$$

Thus $\mathbf{v} \in \mathcal{T}_{\mathcal{A}}(\boldsymbol{\theta}^*)$. Repeat the construction of $\mathbf{v}$ for each $i \in \text{supp}(\boldsymbol{\theta}^*)$, and then the conic combination of these $\mathbf{v}$'s forms $\mathcal{T}_2$. Therefore we have $\mathcal{T}_2 \subseteq \mathcal{T}_{\mathcal{A}}(\boldsymbol{\theta}^*)$, which together with $\mathcal{T}_1 \subseteq \text{cl}(\mathcal{T}_{\mathcal{A}}(\boldsymbol{\theta}^*))$ implies $\mathcal{T}_1 \oplus \mathcal{T}_2 \subseteq \text{cl}(\mathcal{T}_{\mathcal{A}}(\boldsymbol{\theta}^*))$. ∎

## 3 Proof of Theorem 8

**Statement of Theorem:** *For a given $\boldsymbol{\theta}^*$, Algorithm 1 returns a solution to polar operator* (11) *for $\|\cdot\|_k^{sp^*}$.*

*Proof:* The polar operator for 2-$k$ symmetric gauge norm is essentially

$$\mathbf{u}^* = \text{argmax} \langle \mathbf{u}, \boldsymbol{\theta}^* \rangle \quad s.t. \quad \|\mathbf{u}^*\|_{(k)} \leq 1 .$$

As 2-$k$ symmetric gauge norm is sign and permutation invariant, $\mathbf{u}^*$ should conform with the sign and order of $\boldsymbol{\theta}^*$ in order to achieve maxima, i.e., $\langle \mathbf{u}^*, \boldsymbol{\theta}^* \rangle \leq \langle |\mathbf{u}^*|^\downarrow, |\boldsymbol{\theta}^*|^\downarrow \rangle$. W.l.o.g, we assume $\boldsymbol{\theta}^* = |\boldsymbol{\theta}^*|^\downarrow \triangleq \mathbf{z}$. Now we analyze the structure of the solution $\mathbf{u}^*$, whose entries should be nonnegative and sorted in descending order. Assume that $u_k^*$ takes certain fixed but unknown value $\beta$. It is easy to see that the entries in $\mathbf{u}_{k+1:p}^*$ can take the value of $\beta$, as it will always maximize $\langle \mathbf{u}_{k+1:p}^*, \boldsymbol{\theta}_{k+1:p}^* \rangle$ without violating the constraint $\|\mathbf{u}^*\|_{(k)} \leq 1$. Generally we also assume that $\mathbf{u}_{i:k}^*$ take the value of $\beta$ and $u_{i-1}^* > u_i^*$. Then the maximization problem becomes

$$\max_{\mathbf{u}_{1:i-1}, \beta} \langle \mathbf{u}_{1:i-1}, \mathbf{z}_{1:i-1} \rangle + \beta \|\mathbf{z}_{i:p}\|_1$$
$$\text{s.t.} \quad \|\mathbf{u}_{1:i-1}\|_2^2 \leq 1 - (k - i + 1)\beta^2, \ u_j > \beta \text{ for } 1 \leq j < i .$$

Then we let $\mathbf{w} = \mathbf{u}_{1:i-1}$ and introduce the Lagrange multiplier $\boldsymbol{\lambda} \in \mathbb{R}^{i-1}$ and $\alpha \in \mathbb{R}$. Using strong duality, we have the equivalent problem

$$\min_{\boldsymbol{\lambda} \succeq \mathbf{0}, \alpha \geq 0} \max_{\beta, \mathbf{w}} \langle \mathbf{w}, \mathbf{z}_{1:i-1} \rangle + \beta \|\mathbf{z}_{i:p}\|_1 + \langle \boldsymbol{\lambda}, \mathbf{w} - \mathbf{b} \rangle - \alpha((k - i + 1)\beta^2 + \|\mathbf{w}\|_2^2 - 1) ,$$

where $\mathbf{b} = [\beta, \beta, \ldots, \beta]^T \in \mathbb{R}^{i-1}$. By complementary slackness, we know $\boldsymbol{\lambda} = \mathbf{0}$ for the optimal solution if it is feasible. Taking the gradient of the objective function w.r.t $\beta$ and $\mathbf{w}$, we obtain

$$\|\mathbf{z}_{i:p}\|_1 - \sum_i \lambda_i - 2\alpha\beta(k - i + 1) = \|\mathbf{z}_{i:p}\|_1 - 2\alpha\beta(k - i + 1) = 0 \qquad (\text{S.2})$$

$$\mathbf{z}_{1:i-1} + \boldsymbol{\lambda} - 2\alpha\mathbf{w} = \mathbf{z}_{1:i-1} - 2\alpha\mathbf{w} = 0 . \tag{S.3}$$

It is also not difficult to see that the optimal solution will make the constraint $\|\mathbf{u}_{1:i-1}\|_2^2 \le 1 - (k - i + 1)\beta^2$ hold with equality, i.e.,

$$\|\mathbf{w}\|_2^2 = 1 - (k - i + 1)\beta^2 \tag{S.4}$$

Combining the Equation (S.2) (S.3) (S.4), we solve $\beta$ and $\alpha$ and $\mathbf{w}$

$$\beta = \frac{\|\mathbf{z}_{i:p}\|_1}{\sqrt{\|\mathbf{z}_{i:p}\|_1^2(k - i + 1) + \|\mathbf{z}_{1:i-1}\|_2^2(k - i + 1)^2}}, \quad \alpha = \frac{\|\mathbf{z}_{1:i-1}\|_2}{2\sqrt{1 - (k - i + 1)\beta^2}}, \quad \mathbf{w} = \frac{\mathbf{z}_{1:i-1}}{2\alpha},$$

which is essentially the Line 3 in Algorithm 1. As we do not know the $i$ beforehand, we have to check every possible $1 \le i \le k$ to find the one that achieves the maxima without violating the constraint, which corresponds to the loop and if-then statement in Algorithm 1. Since the optimal $\mathbf{w}$ is proportional to $\mathbf{z}_{1:i-1}$, which is sorted in descending order, we only need to ensure $\beta < w_{i-1}$. ∎

## 4 Proof of Theorem 9

**Statement of Theorem:** *For given $s$-sparse $\boldsymbol{\theta}^* \in \mathbb{R}^p$, the Gaussian width $w(\mathcal{C}_k^{sp}(\boldsymbol{\theta}^*))$ and the restricted norm compatibility $\Psi_k^{sp}(\boldsymbol{\theta}^*)$ for a specified $k$ are given by*

$$w(\mathcal{C}_k^{sp}(\boldsymbol{\theta}^*)) \le \begin{cases} \sqrt{p} , & \text{if } s < k \\ \sqrt{\frac{3}{2}s + \frac{2\theta_{\max}^{*2}}{\theta_{\min}^{*2}} s \log\left(\frac{p}{s}\right)} , & \text{if } s = k \\ \sqrt{\frac{3}{2}s + \frac{2\kappa_{\max}^2}{\kappa_{\min}^2} s \log\left(\frac{p}{s}\right)} , & \text{if } s > k \end{cases} , \quad \Psi_k^{sp}(\boldsymbol{\theta}^*) \le \begin{cases} \sqrt{\frac{2p}{k}} , & \text{if } s < k \\ \sqrt{2}(1 + \frac{\theta_{\max}^*}{\theta_{\min}^*}) , & \text{if } s = k \\ (1 + \frac{\kappa_{\max}}{\kappa_{\min}})\sqrt{\frac{2s}{k}} , & \text{if } s > k \end{cases} ,$$
$$\tag{S.5}$$

*where $\theta_{\max}^* = \max_{i \in \mathrm{supp}(\boldsymbol{\theta}^*)} |\theta_i^*|$ and $\theta_{\min}^* = \min_{i \in \mathrm{supp}(\boldsymbol{\theta}^*)} |\theta_i^*|$.*

*Proof:* For $s < k$, we note that $\|\boldsymbol{\theta}^*\|_k^{sp} = \|\boldsymbol{\theta}^*\|_2$, and $\mathbf{u}^*$ can be obtained in a closed-form $\mathbf{u}^* = \frac{\boldsymbol{\theta}^*}{\|\boldsymbol{\theta}^*\|_2}$. Applying Theorem 4, we find that the set $\mathcal{R}$ is empty, and thus the Gaussian width $w(\mathcal{C}_k^{sp}(\boldsymbol{\theta}^*)) = \sqrt{p}$. For $s = k$, $\mathbf{u}^*$ is in closed-form as well,

$$u_i^* = \begin{cases} \frac{\theta_i^*}{\|\boldsymbol{\theta}^*\|_2} , & \text{if } i \in \mathrm{supp}(\boldsymbol{\theta}^*) \\ \frac{|\boldsymbol{\theta}^*|_k^{\downarrow}}{\|\boldsymbol{\theta}^*\|_2} = \frac{\theta_{\min}^*}{\|\boldsymbol{\theta}^*\|_2} , & \text{if otherwise} \end{cases} .$$

In this case, $\mathcal{Q}$ is empty, $\mathcal{R}$ is nonempty, and $|\mathcal{S}| = s = k$. Hence Theorem 4 implies the corresponding Gaussian width, and $\frac{\kappa_{\max}}{\kappa_{\min}} = \frac{\theta_{\max}^*}{\theta_{\min}^*}$. For $s > k$, the closed-form solution is generally unavailable, but we can see from Algorithm 1 that $\beta$ should be nonzero, thus $\mathcal{Q}$ is empty and $\mathcal{R}$ is nonempty, which gives us the corresponding Gaussian width.

Given the fact that $\|\cdot\|_k^{sp} \le \sqrt{2}\max\{\|\cdot\|_2, \frac{\|\cdot\|_1}{\sqrt{k}}\}$ shown in [1], we can choose $\beta_1 = \sqrt{\frac{2}{k}}$ and $\beta_2 = \sqrt{2}$. Base on the analysis above, the restricted norm compatibility for $s \ge k$ directly follows Theorem 5. For $s < k$, we need to compute the unrestricted norm compatibility $\Phi$. As $\|\cdot\|_k^{sp} < \sqrt{2}\max\{\|\cdot\|_2, \frac{\|\cdot\|_1}{\sqrt{k}}\}$, we have

$$\Phi = \sup_{\mathbf{u} \in \mathbb{R}^p} \frac{\|\mathbf{u}\|_k^{sp}}{\|\mathbf{u}\|_2} \le \sup_{\mathbf{u} \in \mathbb{R}^p} \frac{\sqrt{2}\max\{\|\mathbf{u}\|_2, \frac{\|\mathbf{u}\|_1}{\sqrt{k}}\}}{\|\mathbf{u}\|_2} \le \max\{\sqrt{2}, \sqrt{\frac{2p}{k}}\} = \sqrt{\frac{2p}{k}} ,$$

in which we use the inequality $\|\cdot\|_1 \le \sqrt{p}\|\cdot\|_2$. ∎

## References

[1] A. Argyriou, R. Foygel, and N. Srebro. Sparse prediction with the $k$-support norm. In *Advances in Neural Information Processing Systems (NIPS)*, 2012.