[Reviews · NeurIPS 2015]

Submitted by Assigned_Reviewer_1

=== Summary ===

In this paper, the authors propose to bound geometric quantities that appears in statistical error bounds of structured estimation problems with atomic norms.

It was previously shown that the estimation error of Dantzig-like estimator, with random Gaussian design, could be bounded by using three key quantities: the Gaussian width of the unit ball of the regularizer, the Gaussian width of the spherical cap of the tangent cone and the restricted norm compatibility constant. The goal of this article is to provide easy to compute bounds for these three quantities, in the case of sign-invariant atomic norms.

In section 3, the authors introduce upper bounds on the three geometric quantities, which are the main contribution of the paper. The Gaussian width of the unit ball can be bounded by decomposing the set of atoms as the union of simpler sets, for which the Gaussian width are easy to compute. This result already appeared in [14]. The Gaussian width of the spherical cap of the tangent cone can be bounded using a single subgradient of the regularizer, taken at the true parameter vector w^*. Finally, the restricted norm compatibility constant can be bounded using two other "norm compatibility constants" which are simpler to compute (because the set on which they are defined are simpler). The authors show on two examples, the L1 and OWL norms, how these results can easily be applied.

In section 4, the authors provide lower bounds for the two last quantities. The lower bounds does not match the upper bounds, but show that the upper bound are relatively tight, and thus usefull.

In section 5, the bounds obtained in this paper are applied to the k-support norm, improving previously known results.

=== Significance ===

I believe this paper is addressing an important problem, providing interesting and useful new tools for studying sparse estimation problems with atomic norms. The paper is clearly written and well organized. The use of examples helps to understand the results and how to apply those in practice. Finally, the bounds derived in the paper allowed to obtain new results for the k-support norm, showing the relevance of the results.
Summary: I believe this is a well written and interesting paper, providing easy to compute bounds to study the statistical error of Dantzig-like estimator with atomic norms.

Submitted by Assigned_Reviewer_2

This paper studies atomic norms used in structured sparse regression. For a given atomic set A (satisfying a sign-change invariance assumption), the paper bounds the Gaussian width of the unit norm ball, of the tangent cone, and the L2-compatibility number--quantities which are sufficient to characterize the rate of convergence of the estimation problem under certain settings.

The new technique is general. When applied to the L1 norm, both unweighted and weighted, it is shown to match bounds derived from existing techniques. The authors also apply the new technique on k-support norms and derive novel bounds; the new bounds have the interesting implication that k should be under-specified in practice.

The unit norm ball analysis builds on a lemma from [14] and does not seem a significantly novel contribution. The tangent cone and the compatibility analyses look nontrivial. They build upon a simple but powerful observation that the tangent cone of an atomic norm can be upper bounded by the tangent cone of a weighted L1 norm (Lemma 3).

My one suggestion is that the authors perform numerical simulations with k-support norms and test whether the bound matches actual behavior. It would be interesting to see whether over-specifying k would hurt recovery.

I find the paper interesting and the proof relatively easy to follow. The paper is well-organized and the results clearly stated and explained.

[14] An inequality with application to structured sparsity and multi-task dictionary learning.
Summary: This paper gives a new method of bounding the Gaussian width of various sets important for deriving the rates of consistency of predictors with structured sparsity penalties. The new method seems innovative and gives a novel convergence rate for regressions with k-support norms.

Submitted by Assigned_Reviewer_3

This paper is about sample complexity and error bounds for Dantzig-type estimators in the context of structured regression. A central ingredient of such bounds are geometric properties like the the Gaussian width of the norm ball and similar related quantities. For the class of atomic norms that are invariant under sign changes, novel bounds of these geometric properties are provided, which can be easily computed in practice. Further, tightness of the bounds is analyzed by showing that these upper bounds are close to the corresponding lower bounds.

The main focus of this work concerns a topic which is definitely relevant and interesting to the Machine Learning community. The papers is written in a clear and transparent way. In my opinion, the bounds obtained are both of theoretical and practical interest. However, I have to admit that I did not fully understood some technical details like the proof of Theorem 7.

Summary: An interesting theory paper about a highly relevant topic. Well written.

Submitted by Assigned_Reviewer_4

The authors consider sparse linear model estimation with atomic norms and provide some bounds on Gaussian widths arising in their analysis. As a special case, the authors derive bounds for l1 and ordered wighted l1 norm and provides some analysis on the recently proposed k-support norms.

The atomic norm is attractive for its generality to capture a broad class of linear inverse problems [Chandrasekaran et al, 2012 FOCM]. The present paper, however, focuses only on sparse linear models and the results, except for the recently proposed k-support norm, are already available. The proof techniques for calculating Gaussian widths are also fairly standard, although the dual norm calculation for k-support norm analysis is new.

Is this technique generalizable to other instances of atomic norm analysis, e.g., nuclear norm penalized estimation of low-rank matrices?

The paper is well-written, the proofs are clear and easy to follow.

Summary: The paper is well-written and there is no gap in the mathematical proofs. However the scope of the problem is narrow: instead of providing analysis for general atomic norm analysis [Chandrasekaran et al, 2012 FOCM] unifying in a variety of linear inverse problems, the authors only consider sparse linear regression for which the key results are already available.

Author Feedback
Author rebuttal: Assigned Reviewer 1:
We thank the reviewer for the encouraging comments. First we agree that the unit norm ball analysis is mainly built on [14], but we demonstrate that the application of the inequality needs care, e.g., a naive application to OWL will result in a O(\sqrt{p}) bound since there are 3^p-1 atoms, while applying the inequality hierarchically yields the desired sharp bound. Second, the k-support norm analysis actually explains prior experimental results, e.g., figure 2(c) in [10], where the L2 error starts increasing when k crosses the true sparsity. We will make this aspect more clear.

Assigned Reviewer 2:
We thank the reviewer for encouraging comments. We will highlight the main contributions in more detail in an extended version of the work - thanks for the suggestion.

Assigned Reviewer 5:
We thank the reviewer for the encouraging comments. It is true that [Chandrasekaran et al, 2012] focuses on both vector and matrix scenarios. However, the Gaussian width of the tangent cone is calculated individually for certain norms, most of which are decomposable. Our work establishes such a general bound for any atomic norm as long as the norm is sign-invariant, which is true for commonly used norms. We also have the restricted norm compatibility analysis, which is not needed for the estimator in [Chandrasekaran et al, 2012]. Regarding (structured) low-rank matrix estimation, our result can be extended to unitarily invariant matrix norms (see "The Convex Analysis of Unitarily Invariant Matrix Functions" by A.S. Lewis, 1995), which already include nuclear, Frobenius, spectral norms, and the spectral k-support norm [15]. Some technical details will be a little different from vector setting, e.g., the notion of sign-change invariance. We can add a remark and the reference to Lewis'95 to highlight the generality of our results. For other statistical models beyond linear, e.g., generalized linear models, the error bounds also involve these geometric measures (see [3]), so our results apply.

Assigned Reviewer 6:
We thank the reviewer for the comments. We want to clarify the key differences between our paper and existing work such as [10]. The existing work primarily focuses on deriving error bounds such as equation (5) using geometric measures (see [3, 9, 10]). However, there is no general and easy way of bounding these geometric measures. Although some results are available for specific decomposable norms such as L1, results are unknown for general non-decomposable norms. Our work introduces a general approach to bounding the geometric measures, and is applicable to any atomic norm as long as they are sign invariant, which is true for commonly used norms. To the best of our knowledge, our main idea of constructing a superset cone induced by weighted L1 semi-norm is new. It is different from the techniques used in [10], which yields a weaker bound for k-support norm. For k-support norm, we also provide a new algorithm to compute the polar operator, which can be useful in the evaluation of k-support norm, or certain optimization frameworks that require solving polar operator (see [24]). Moreover, the geometric measures we analyzed also appear in the error bounds of other statistical estimation problems (see [3]), e.g., generalized linear models, so our results apply as well.

Assigned Reviewer 7:
We thank the reviewer for the encouraging comments. We agree that some of the proofs were rather brief (due to the page limit) and we skipped some relevant results from [9] that we used in the proofs. We plan to do an extended version with self-contained proofs and sufficient details on the relevant background, e.g., the notion of tangent/normal cone and statistical dimension.

Assigned Reviewer 8:
We thank the reviewer for the encouraging comments. The reviewer has summarized the work very clearly as part of the review, and we appreciate this.